# Functional Characterization of Pheromone Receptors in the Beet Webworm, *Loxostege sticticalis* (Lepidoptera: Pyralidae)

**DOI:** 10.3390/insects14070584

**Published:** 2023-06-27

**Authors:** Yu Zhang, Hai-Bin Han, Yan-Yan Li, Lin-Bo Xu, Li-Fen Hao, Hui Wang, Wen-He Wang, Shu-Jing Gao, Ke-Jian Lin

**Affiliations:** 1Key Laboratory of Biohazard Monitoring, Green Prevention and Control for Artificial Grassland, Ministry of Agriculture and Rural Affairs, Institute of Grassland Research of Chinese Academy of Agricultural Sciences, Hohhot 010010, China; zhangyu11@caas.cn (Y.Z.); hhb.25@163.com (H.-B.H.); xulinbo@caas.cn (L.-B.X.); haolifen@caas.cn (L.-F.H.); wh1978xx@163.com (H.W.); 2Research Center for Grassland Entomology, Inner Mongolia Agricultural University, Hohhot 010020, China; liyanyan@imau.edu.cn; 3Forest Farm of Baichengzi of Alukeerqin Banner, Chifeng 024000, China; aqbczlcdzb@163.com

**Keywords:** *Loxostege sticticalis*, pheromone receptors, electroantennography, signal sensillum recordings, *Xenopus* oocytes

## Abstract

**Simple Summary:**

The moth *Loxostege sticticalis* (Lepidoptera: Pyralidae) is a typical migrant insect pest found in North America, Eastern Europe, and Asia. The natural pheromone compounds of adult moths are involved in courtship and mating behavior. In this article, we examine the peripheral mechanisms of sex communication, demonstrating that the pheromone receptor (PR) LstiPR2 responds specifically to the major sex pheromone compound *E*11-14:OAc, which results in the activation of the “a” neuron in sensilla trichodea. The other four LstiPRs showed no response to any sex pheromone compounds. This study will contribute to a new, environmentally friendly strategy for integrated pest management of *L. sticticalis*.

**Abstract:**

Lepidopteran insects mainly rely on sex pheromones to complete sexual communications. Pheromone receptors (PRs) are expressed on the olfactory receptor neurons (ORNs) of the sensilla trichodea and play an essential role in sexual communication. Despite extensive investigations into the mechanisms of peripheral recognition of sex pheromones in Lepidoptera, knowledge about these mechanisms in *L. sticticalis* remains limited. In this study, five candidate LstiPRs were analyzed in a phylogenetic tree with those of other Lepidopteran insects. Electroantennography (EAG) assays showed that the major sex pheromone component *E*11-14:OAc elicited a stronger antennal response than other compounds in male moths. Moreover, two types of neurons in sensilla trichodea were classified by single sensillum recordings, of which the “a” neuron specifically responded to *E*11-14:OAc. Five candidate PRs were functionally assayed by the heterologous expression system of *Xenopus* oocytes, and LstiPR2 responded to the major sex pheromone *E*11-14:OAc. Our findings suggest that LstiPR2 is a PR sensitive to *L. sticticalis*’s major sex pheromone compound, *E*11-14:OAc. Furthermore, this study offers valuable insights into the sexual communication behavior of *L. sticticalis*, forming a foundation for further analysis of the species’ central nervous system.

## 1. Introduction

Insects have a sophisticated olfactory system to detect and distinguish a wide range of environmental odorants with high sensitivity and specificity [1,2]. Insects rely on the olfactory system to locate habitats and identify mating partners. Sex pheromones emitted from conspecifics are crucial for courtship and mating [3,4]. The sex pheromones of Lepidoptera species, composed of two or more components, are biosynthesized and released by the female sex gland to attract male moths [5]. Since the first insect sex pheromone ((*Z*,*E*)-10,12-hexadecadienol) was identified in *Bombyx mori* [6], increasing numbers of insect sex pheromone compounds have been identified and introduced into agriculture to replace conventional pest control agents [7,8,9,10,11,12].

In Lepidoptera, the primary olfactory organs for recognizing sex pheromones are the antennae. The surface of the antennae is covered with hair-like protrusions called sensilla. Many types of sensilla have been identified, such as trichodea, basiconica, coeloconica, chaetica, squamiformia, styloconica, and Böhm’s bristles [13,14]. The functions of different types of sensilla are diverse, including chemosensation, hygrosensation, theromosensation, and mechanosensation [15,16]. Among these sensilla types, the sensilla trichodea is associated with pheromone detection [17,18,19,20,21]. A typical feature of the sensilla trichodea is the presence of pores on the surface [20,22]. In male antennae, the hairs participating in pheromone sense are common in most moths, such as *B. mori*, *Manduca sexta*, *Agrotis ipsilon*, and *Helicoverpa armigera* [23].

Sexual communication involves several olfactory proteins that detect, recognize, and degrade peripheral sex pheromones. Odorant binding proteins (OBPs), chemosensory proteins (CSPs), odorant receptors (ORs), ionotropic receptors (IRs), sensory neuron membrane proteins (SNMPs), and odorant degrading enzymes (ODEs) play individual and cooperative roles in these processes [17,24]. Each OR is generally co-expressed with the OR co-receptor (Orco), facilitating chemical signal recognition. This process transforms the chemical signals into electrical signals, regulating chemical communication in insects [4,25]. Pheromone receptors (PRs) are the main ligand gated-ion channels used to bind to pheromones, which are members of the ORs superfamily [24].

Recent studies have reported the functions of insect PRs using *Xenopus* oocytes [26,27]. For instance, the AdisOR1 from *Athetis dissimilis* exhibited a strong response to the sex pheromone compounds (*Z*)-9-tetradecenol (*Z*9-14:OH) and (*Z*,*E*)-9,12-tetradecadien-1-ol (*Z*9, *E*12-14:OH) [28]. Similarly, AlepOR3 from *Athetis lepigone* demonstrated high sensitivity to the sex pheromone (*Z*)-7-dodecenyl acetate (*Z*7-12:OAc) [29]. DpunPR45 from *Dendrolimus punctatus* showed a broad response to sex pheromone compounds [30]. The response profile of olfactory receptor neurons (ORNs) housed in the sensilla, which is determined by expressed PRs, is an essential molecular element in the peripheral olfactory process in insects [27,31]. PRs mainly participate in the sexual communication process of moths [32]. Therefore, PRs play a crucial role in detecting pheromone compounds, and it is necessary to research the functions of PRs in depth.

*Loxostege sticticalis* (Lepidoptera: Pyralidae), commonly known as the beet webworm, is a devastating pest causing substantial ecological and economic damage to numerous crops across North America, Eastern Europe, and Asia [33,34,35]. *L. sticticalis* was added to the National Class I list of crop insect pests in 2020 by the Chinese Ministry of Agriculture and Rural Affairs. Like most moths, the male moth of *L. sticticalis* engages in courtship and mating by detecting the sex pheromone components emitted by female moths [36,37]. Sex pheromones in Lepidoptera are primarily classified into three types—Type 0, Type I, and Type II—based on their chemical structures [38]. The sex pheromone components of *L. sticticalis* belong to Type I pheromones, consisting of straight-chain alcohols, aldehydes, and acetates with 10–18 carbon atoms. The compounds 11€-tetradecenyl acetate (*E*11-14:OAc), 11(*E*)-tetradecenal (*E*11-14:Ald), *E*-11-tetradecenol (*E*11-14:OH), 1-tetradecanol (14:OH), tetradecyl acetate (14:OAc), and 1-dodecanol (12:OH) are the sex pheromone components of *L. sticticalis* [38,39,40]. A mixture of (*E*11-14:OAc):(*E*11-14:Ald) and (*E*11-14:OAc):(*E*11-14:Ald):(*E*11-14:OH) at a ratio of 1:1 and 5:3:12, respectively, was found to be optimal lure rations for *L. sticticalis* male moths in wind tunnel assays and field assays [39].

While the technique of RNA-Seq [34] has identified the five candidate *LstiPRs* involved in pheromone detection in *L. sticticalis*, the function of these genes remains unknown. In this study, we cloned the full-length sequences of five candidate *LstiPRs* and *LstiORco* from the antennae of male specimens. The functions of the candidate LstiPRs were characterized using heterologous expression in the *Xenopus* oocyte system with a two-electrode voltage clamp, and single sensillum recordings (SSRs) were performed to analyze the function of olfactory receptor neurons in response to sex pheromone compounds.

## 2. Materials and Methods

### 2.1. Insects

In June 2021, over a thousand larvae of *L. sticticalis*, ranging from two to five instars, were collected from Hohhot, Inner Mongolia, China (40°82′17″ N, 111°71′58″ E). The larvae were reared with fresh *Chenopodium album* at a constant temperature of 22 ± 1 °C, a photoperiod of 16 h of light and 8 h of darkness, and a relative humidity of 75 ± 5%. The last-instar larvae were transferred to a box containing clean, sandy soil at 15% humidity to facilitate pupation. Newly emerged moths were immediately separated by sex and kept in 500 mL beakers with a 5% honey solution.

### 2.2. RNA Extraction and cDNA Synthesis

The antennae were isolated from three-day-old male adults, immediately snap-frozen in liquid nitrogen, and stored at −80 °C for subsequent use. The total RNA of male antennae (30 pairs) was extracted using TRIZOL^®^ Reagent (Invitrogen) according to the manufacturer’s instructions. The quality and quantity of the RNA were assessed by agarose gel electrophoresis (AGE) and NANODROP 2000 (ThermoFisher, Waltham, MA, USA), respectively. The first-strand cDNA was synthesized using the RevertAid First Strand cDNA Synthesis Kit (ThermoFisher, Waltham, MA, USA).

### 2.3. Phylogenetic Analysis

Phylogenetic trees were constructed using amino acid sequences of candidate genes from *LstiPRs*, *LstiORco*, from Lepidoptera species, including *B. mori* [41], *Cnaphalocrocis medinalis* [42], *Ostrinia furnacalis* [43], and *L. sticticalis* [34]. The amino acid sequences were aligned using Multiple Alignments using Fast Fourier Transform (MAFFT; https://www.ebi.ac.uk/Tools/msa/mafft/ (accessed on 20 April 2022)) with default parameters, and the phylogenetic tree was constructed using the Jones Taylor Thornton (JTT) method in RAxML version 8 with 1000 bootstrap tests [44].

### 2.4. Cloning Full-Length Sequence of LstiPRs

Following transcriptome analysis and polymerase chain reaction (PCR), the full-length cDNA sequences of five candidate *LstiPRs* were obtained. The *LstiPR*-specific primer pairs were designed using Primer Premier 5.0 software (PRIMER Biosoft, Palo Alto, CA, USA) (Table 1). The volumes of the amplification reaction were 50 μL, including 2 μL male antennal cDNA (1 μg μL^−1^), 2 μL (each) of the primer pairs (10 μM), 19 μL ddH_2_O, and 25 μL 2 × PrimerSTAR^®^ HS DNA Polymerase (TaKaRa, Japan). PCR conditions were: pre-denaturation at 98 °C for 45 s, followed by 36 cycles of 98 °C for 10 s, 60 °C for 15 s, and 72 °C for 2 min. The final extension was at 72 °C for 10 min. Each *LstiPR* was initially ligated into the blunt cloning vector of the pEASY^®^-Blunt Simple Cloning Kit (TransGen Biotech, China) according to the manufacturer’s instructions. Subsequently, the *LstiPRs* were cloned into the pT7Ts-vector (Invitrogen, Waltham, MA, USA) by seamless cloning technology, according to the instructions of the ClonExpress^®^ II One Step Cloning Kit (Vazyme, Nanjing, China).

### 2.5. Pheromone Components

Six sex pheromone compounds have been identified from *L. sticticalis* previously, including (*E*)-11-tetradecenyl acetate (*E*11-14:OAc), (*E*)-11-tetradecenal (*E*11-14:Ald), (*E*)-11-tetradecen-1-ol (*E*11-14:OH), 1-tetradecanol (14:OH), tetradecyl acetate (14:OAc), and 1-dodecanol (12:OH). The pheromone compounds were purchased from Changzhou Nimrod Biotech Inc. (Changzhou, China) (Table 2). Stock solutions of the compounds were prepared in dimethyl sulfoxide at a concentration of 1 M. For the *Xenopus* oocyte system experiments, 1 × Ringer’s buffer (96 mM NaCl, 2 mM KCl, 5 mM MgCl_2_, 0.8 mM CaCl_2,_ and 5 mM HEPES, pH 7.6) was used to dilute the stock solutions into working solutions.

### 2.6. Electroantennography Responses

Electroantennography (EAG) was used to evaluate the response of the antennae of three-day-old male adults to six pheromone compounds of females in *L. sticticalis*. Briefly, the pheromone compounds were dissolved in paraffin oil and diluted to 10 μg μL^–1^. A filter paper (0.5 cm × 5 cm) was loaded with 10 μL pheromone compound using a Pasteur pipette, and the duration of stimulation was set to 0.8 s with an interval of 3 min. The negative control was 10 μL paraffin oil only. Fifteen to twenty male antennae were stimulated with each pheromone compound, and the difference in electric signal values before and after stimulation was compared. The signals were amplified by a 10 × AC/DC preamplifier (Syntech) and recorded by Syntech EAG software (EAGPro 2.0). The values were normalized to the negative control.

### 2.7. Single Sensillum Recordings

Signal sensillum recordings (SSRs) revealed the peripheral neural coding of *L. sticticalis* males detecting pheromones. Three-day-old male antennae of sensilla trichodea were utilized to record responses to pheromone compounds. A single male moth was immobilized inside a 1 mL plastic pipette with the narrow end trimmed off. The moth was delicately pushed into the plastic tip, leaving one antenna and head outside, and secured by dental wax. The tungsten microelectrodes were sharpened with a solution of KNO_2_ (40%). The recording electrode was inserted into the antenna’s sensilla, while the reference electrode was inserted into the opposite eye. Each trial was performed to test the responses to 10 μL of pheromone compounds at a concentration of 10 μg μL^−1^ that was dripped onto a piece of filter paper (0.5 cm × 5 cm) and inserted into a Pasteur pipette. The continuous purified and humidified air flow stream was set at 1.2 L min^−1^ with a 0.3 s stimulus air pulse. A total of eight sensilla trichodea were recorded successfully. The recorded signal was amplified through a 10 × AC/DC preamplifier (Syntech) and recorded by a data acquisition controller (10 s, starting 1 s before stimulation) (IDAC-4, Syntech, Kirchzarten, Germany). The software Autospike (Syntech, Kirchzarten, Germany) was used for digitizing and displaying action potentials on a computer screen. The response of the sensilla trichodea was measured by the change in action potential frequency relative to the frequency of spontaneous action potentials (spikes s^−1^) before and after stimulation [45].

### 2.8. Xenopus Oocyte System and Two-Voltage Clamp Recordings

The cDNA sequences of candidate pheromone receptors were cloned, and cRNA was synthesized according to the mMESSAGE mMACHINETM T7 Kit (Thermo Fisher Scientific) instructions. A mixture of each candidate LstiPR and LstiORco cRNA (1:1) was injected into *Xenopus laevis* oocytes (stageV/VI), which were treated with 2 mg mL^−1^ collagenase I in Washing Buffer (96 mM NaCl, 2 mM KCl, 5 mM MgCl_2,_ and 5 mM HEPES, pH 7.6) for about 1 h at room temperature and then incubated overnight in Barth solution (96 mM NaCl, 2 mM KCl, 0.8 mM CaCl_2_, 5 mM MgCl_2_, 5 mM HEPES, 50 mg/mL tetracycline, 5% horse serum, 100 mg ml^−1^ streptomycin, and 550 mg ml^−1^ sodium pyruvate, pH 7.6). After incubating for three days at 18 °C, the two-electrode voltage clamp system was employed to measure the cellular current elicited by stimulation with various pheromones at a concentration of 10^−4^ M. Dose-response curves were performed using a concentration range of 10^−7^ to 5 × 10^−3^ M for each pheromone compound, testing at least six oocyte replicates. The electric signal was amplified using an OC-725C oocyte clamp (Warner Instruments, Hamden, CT, USA) at a holding potential of −80 mV. The data were analyzed with Digidata 1440A and pCLAMP 10.0 software (Axon Instruments Inc., Union City, CA, USA).

### 2.9. Data Analysis

Data analysis were performed using SPSS 17.0 (IBM Corp., Chicago, IL, USA), and graphs were created with GraphPad Prism 7.0 (GraphPad Software Inc., Boston, CA, USA). The electrophysiological values (currents and spikes s^−1^) were presented as the mean ± SEM, and the responses of LstiORco/LstiPRx to tested pheromone compounds and single sensillum recording data were analyzed using a one-way nested analysis of variance, followed by a Tukey’s post hoc test (*p* < 0.05) (IBM, Endicott, NY, USA).

## 3. Results

### 3.1. Gene Cloning and Phylogenetic Analysis

The full-length coding regions of the five *LstiPRs* were amplified, and the complete open reading frames of *LstiPR1*, *LstiPR2*, *LstiPR3*, *LstiPR4*, and *LstiPR5* were found to contain 978 bp, 1263 bp, 1086 bp, 1308 bp, and 1038 bp, encoding polypeptides of 326, 421, 362, 436, and 346 amino acids, respectively. The pairwise sequence identities among these five *LstiPRs* ranged from 29.91% (*LstiPR1*/*LstiPR2*) to 60.80% (*LstiPR2*/*LstiPR5*) (Table 3).

A phylogenetic tree was constructed from candidate *LstiORs* together with other ORs from *B. mori* [41], *C. medinalis* [42], and *O. furnacalis* [43] based on maximum likelihood (Figure 1). A clade composed of highly conserved *ORco* genes from *L. sticticalis,* and the three other species was evident. We observed that the LstiORs share high homology with those of *O. furnacalis*, and LstiORco is part of the widely expressed and highly conserved ORco family in insects. The five LstiPRs were clustered into the PR clade, which was easily distinguishable from other Lepidopteran species due to their high degree of similarity (purple shaded region in Figure 1).

### 3.2. Electroantennogram Assays of Pheromone Compounds in L. sticticalis Males

The antennal EAG response of *L. sticticalis* males to six pheromone compounds was evaluated. The results indicate that *E*11-14:OAc, at 10 μg μL^–1^, elicited the strongest response, which was significantly higher than responses to the other pheromones (*p* < 0.05, Figure 2). The pheromone component, 14:OH, elicited a moderate EAG response (Figure 2). However, the other four pheromone components (12:OH, 14:OAc, *E*11-14:OH, and *E*11-14:Ald) elicited weak EAG responses (Figure 2).

### 3.3. The Responses of the Sensilla Trichodea ORNs of L. sticticalis Males to Pheromones

We recorded responses of sensilla trichodea to pheromones from various positions on the antennae of male *L. sticticalis*. The results revealed that each sensillum housed two olfactory neurons: one large-spiking and one small-spiking ORN. These were differentiated by their spike amplitudes in single sensillum recordings (Figure 3A). Among the six pheromone components, *E*11-14:OAc elicited a specific response at a concentration of 10 µg μL^–1^ to the “a” ORN (Figure 3B). However, the remaining five pheromone components did not induce any changes in the number of spikes (Figure 3B,C).

### 3.4. Functional Reconstitution of LstiORco and LstiPRx in Heterologous Cells

The *Xenopus* oocyte system recorded the responses of the five candidate LstiPRs to six pheromone compounds. The results show that LstiPR2 was mainly responsive to *E*11-14:OAc, the main sex pheromone of *L. sticticalis*, with a current value of 702.93 ± 100.86 nA (Figure 4A). In dose-response experiments, LstiPR2/LstiORco was sensitive to *E*11-14:OAc at a low threshold of the response ranging from 310.57 ± 26.84 nA (10^−7^ M) to 561.167 ± 77.56 nA (5 × 10^−4^ M), with an EC_50_ value of 1.249 × 10^−5^ M (Figure 4B). The other four receptors (LstiPR1, LstiPR3, LstiPR4, and LstiPR5) showed no response to the pheromone compounds of *L. sticticalis* (Figure 4C).

## 4. Discussion

Many studies have shown that PRs are vital in insects’ sexual communication [32]. Previous studies identified the pheromone compounds and the candidate PRs of *L. sticticalis* [34,39], but they provided limited characterization and functional analysis of these LstiPRs. To further investigate their roles in sexual communication in *L. sticticalis*, we identified five candidate LstiPRs based on sequence homology and phylogenetic analysis [26]. According to the phylogenetic tree, five candidate LstiPRs clustered into the PRs clade due to their high homology. Therefore, we hypothesized that these five candidate LstiPRs could potentially detect sex pheromones in *L. sticticalis* based on the features of PRs in other moths [46,47].

Sexual communication is essential for courtship and mating behaviors in insects, especially in Lepidoptera [4,37,48]. In general, antennae function as the primary olfactory organs involved in courtship rituals [1,49]. Thus, the antennal responses of male moths to these pheromones were measured by EAG assays to identify the active sex pheromone compounds. Among the six pheromone compounds tested, *E*11-14:OAc elicited the highest EAG responses. Consistent with previous studies, *E*11-14:OAc had an induction effect on the entire wind tunnel process, including flight, upwind, approach, and landing [39,40]. However, the other five pheromone components showed weak EAG responses in males, which may be related to the low numbers of sensilla on the antennae of males [13,15,50]. Many studies have shown that PRs are expressed on the ORNs of dendrites of the sensilla trichodea, which primarily sense pheromones [51,52]. Thus, sensilla trichodea were classified using single sensillum recordings to further characterize the functional types, interpreting the recognition of olfactory sensory neurons as pheromone compounds in *L. sticticalis*. The results show that two neurons, “a” and “b,” were identified in the sensilla trichodea of *L. sticticalis*. ORN “a” in sensilla trichodea was activated by *E*11-14:OAc. Usually, two or three neurons are housed in the sensilla trichodea in Lepidoptera [53]. For instance, in *Spodoptera frugiperda* male’s antennae, two neurons are housed in Type I and III sensilla trichodea, and three are housed in the Type II sensilla trichodea. In addition, *E*11-14:OAc was the main pheromone compound in some moths, such as *O. furnacalis* and *Mythimna loreyi* [46,53]. Our results are consistent with previous studies, namely that the sensilla primarily detected the pheromone compounds [53]. Interestingly, the large-spiking ORN in the male *L. sticticalis* trichoid sensilla was activated by *E*11-14:OAc, also found in *S. frugiperda* and *M. loreyi* [10,53]. However, the remaining pheromone compounds did not induce any response in the sensilla trichodea ORNs, leading us to speculate that other types of antennal sensilla might be responsible for detecting these compounds. Previous studies have shown that intra-specific pheromone compounds can also be sensed by the ORNs of sensilla auricillicum [54]. From the above two experimental results, some discrepancies are clear. EAG results showed that all tested pheromones could activate the male’s antennae response, whereas SSR results indicated that only *E*11-14:OAc could elicit the male’s antennae response. The reason for this discrepancy is that EAG tested the response of the entire moth antennae, whereas SSR assays only tested the sensilla trichodea type. Thus, we speculated that the 14:OH might be recognized by other types of sensilla [14].

To further advance our understanding of the sexual communication process, it is imperative to elucidate the function of PRs, as they play a crucial role in detecting pheromone components predominantly by olfactory genes [11,36]. Currently, *Xenopus* oocytes, as a heterologous expression system in vitro, have been used for studying many species, such as *Epiphyas postvittana* [11], *Heliothis virescens* [55], and *O. furnacalis* [46]. Therefore, we assayed the response of five candidate LstiPRs from *L. sticticalis* to known pheromone compounds by using the *Xenopus* oocyte system and found that LstiPR2 was elicited explicitly by the pheromone component *E*11-14:OAc, while the other four candidate LstiPRs (LstiPR1, LstiPR3, LstiPR4, and LstiPR5) were not activated by any test pheromone compounds. The results show that LstiPRs exhibited a narrow tuning to the sex pheromone chemical signal from female *L. sticticalis*, in contrast to the broadly tuned PRs identified in *O. furnacalis* [46]. These results differ from most studies conducted on other moth species. The possible reason may be that when the PRs are expressed in vivo, other olfactory elements must complete the whole process, such as OBPS, CSPs, and SNMPs [56]. Furthermore, the non-responding candidate LstiPRs (LstiPR1, LstiPR3, LstiPR4, and LstiPR5) might be able to detect host volatiles or other types of odors, as previously reported for *Cydia pomonella,* where CpomOR3 was elicited by a volatile pear ester [57].

## 5. Conclusions

In conclusion, our results provide robust molecular and electrophysiological evidence that *E*11-14:OAc is the major pheromone component in *L. sticticalis*. Interestingly, the pheromone component *E*11-14:OAc exists in the sex pheromone components of many other species, such as *Ostrina* spp. [46], *Proeulia auraria* [58], and *Spodoptera litura* [12]. Therefore, further investigation into the molecular mechanisms that differentiate between inter- and intra-specific pheromone components in *L. sticticalis* is required, which is of great significance for optimizing the current pheromone lure.

## Figures and Tables

**Figure 1 insects-14-00584-f001:**
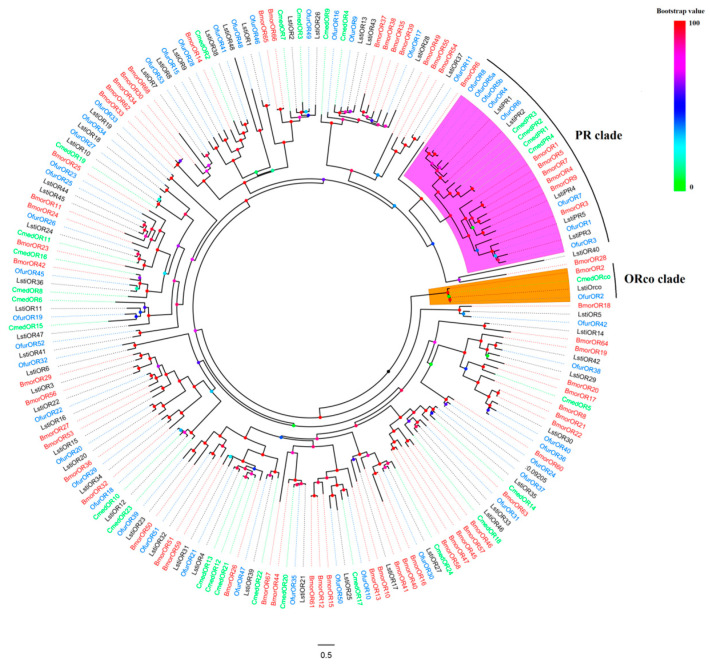
Phylogenetic relationship between odorant receptors (ORs) of *L. sticticalis* and other species. The sequences of various ORs were aligned using MAFFT with default parameters, and the tree was constructed by RAxML 8.0 with a bootstrap of 1000 replicates. The clade of the purple region indicates pheromone receptors (PRs), and the clade of the orange region indicates OR co-receptors (ORco).

**Figure 2 insects-14-00584-f002:**
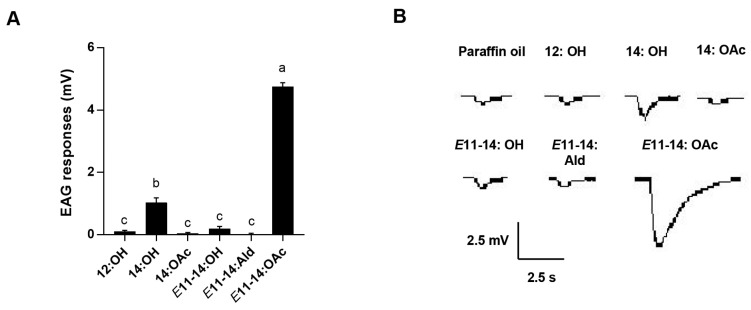
Electroantennogram (EAG) of responses of *L. sticticalis* males to pheromone compounds. (**A**) Responses of males to six pheromone compounds. Error bars represent ± SEM (*n* = 15–20). Lowercase letters indicate significant differences among tested pheromone compounds (*p* < 0.05). (**B**) Tracings of the response of male *L. sticticalis* to six different pheromone compounds.

**Figure 3 insects-14-00584-f003:**
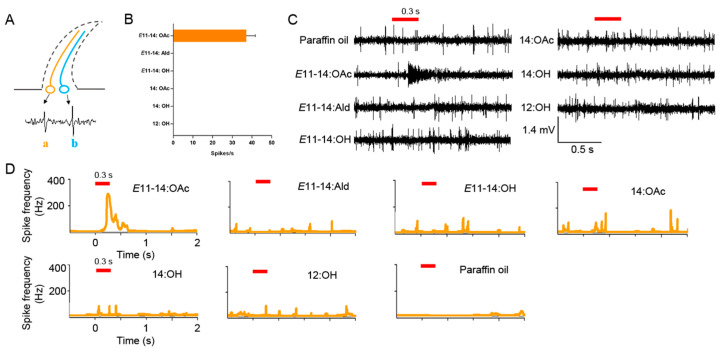
The responses of sensilla trichodea in the antennae of *L. sticticalis* males to pheromone components. (**A**) Distinct ORNs housed in the sensilla trichodea. The order and color of neurons are randomly chosen. “a” and “b” represent the large- and small-spiking ORN, respectively. (**B**) The response profile of sensilla trichoidea. The error bar represents ± SEM (n = 8). (**C**) The response traces of signal sensilla recordings of the sensilla trichodea of *L. sticticalis* males to pheromone components. The red stub line represents stimulus air pulse. (**D**) Peristimulus time histogram (PSTH) showing the responses of ORN “a” (orange) to six odor stimuli and control. The red stub line represents stimulus air pulse.

**Figure 4 insects-14-00584-f004:**
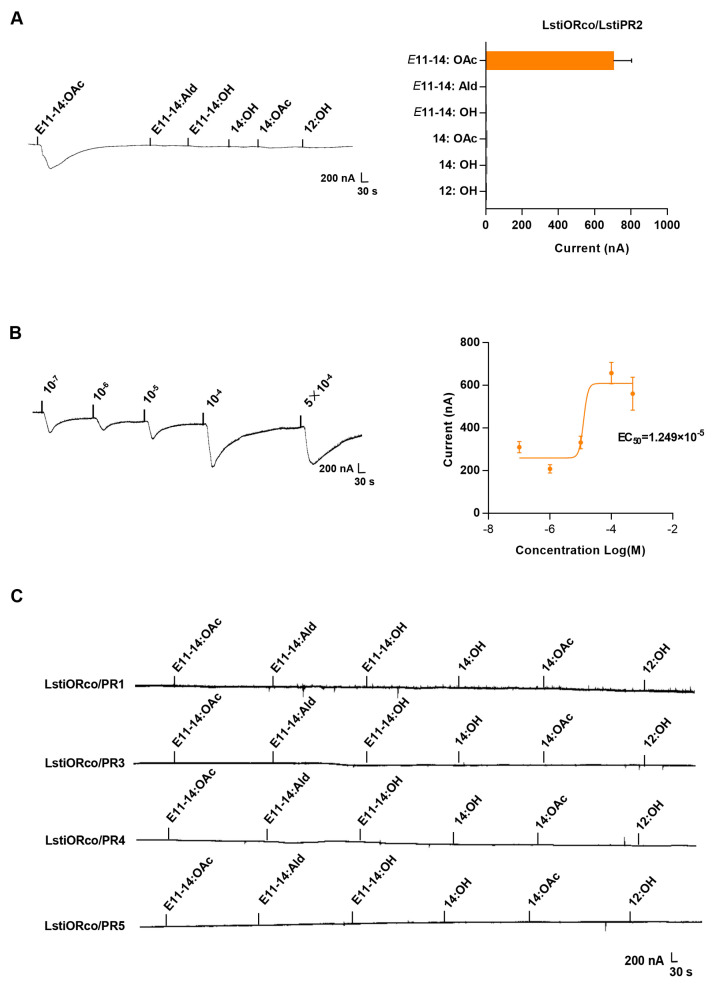
Stimulation of LstiORco/LstiPRx with pheromone compounds in *Xenopus* oocytes. All LstiPRs were co-expressed with LstiORco. The concentration of pheromone compounds was 10^−4^ M, and stimulation was for 15 s. (**A**) Response profile of LstiORco/LstiPR2 in the *Xenopus* oocyte system stimulated with pheromone compounds. Error bars indicate ± SEM (*n* = 6–10). (**B**) Dose-response curve of LstiORco/LstiPR2 in *Xenopus* oocytes stimulated with *E*11-14:OAc. (**C**) Lack of response of LstiPR1, LstiPR3, LstiPR4, and LstiPR5 to different pheromone compounds.

**Table 1 insects-14-00584-t001:** Primers used in this study.

Gene Name	Forward	Reverse
*LstiORco*	atcactagt*gggccc*GCCACCTTCAAGATGATGACCAAAGTGAAA (*ApaI*)	ctagtcagtc*gcggccgc*TCATCTACTTCAGTTGCACCAACA (*NotI*)
*LstiPR1*	atcactagt*gggccc*GCCACCTGGTTATTTCGGTATGAACTCTCTG (*ApaI*)	ctagtcagtc*gcggccgc*TTAATTTAAAGTGTTTGTAAGAATGCATA (*NotI*)
*LstiPR2*	atcactagt*gggccc*GCCACCAAATCCCTCAAAATGAAGAATAAATC (*ApaI*)	ctagtcagtc*gcggccgc*CACTATTCTCCCATCGTTTGCA (*NotI*)
*LstiPR3*	atcactagt*gggccc*GAATCTACGAAAATGTTTAAAATATGCT (*ApaI*)	ctagtcagtc*gcggccgc*GCGATGTCAATGTTCACTACTTCC (*NotI*)
*LstiPR4*	atcactagt*gggccc*GCCACCTTTATGCATAAACTCAGAATGTTTTTAA (*ApaI*)	ctagtcagtc*gcggccgc*GTTTAATCTTTAGTTGCGAAAGTTTG (*NotI*)
*LstiPR5*	atcactagt*gggccc*GCCACCTTCCGCGATGTAAATTACCG (*ApaI*)	ctagtcagtc*gcggccgc*AGTCTCGCTCTGAGTACCGAGAA (*NotI*)

The lowercase letters represent the homology arm, and the italicized lowercase letters represent restriction sites. The names of the restriction enzymes used are indicated in parentheses.

**Table 2 insects-14-00584-t002:** The pheromone compounds used in this study.

Chemicals	CAS Number	Purity (%)
(*E*)-11-tetradecenyl acetate (*E*11-14:OAc)	33189-72-9	>90
(*E*)-11-tetradecenal (*E*11-14:Ald)	35746-21-5	>90
(*E*)-11-tetradecen-1-ol (*E*11-14:OH)	35153-53-8	>95
1-tetradecanol (14:OH)	112-72-1	>97
tetradecyl acetate (14:OAc)	638-59-5	>97
1-dodecanol (12:OH)	112-53-8	>99

**Table 3 insects-14-00584-t003:** Pairwise sequence identities among five *LstiPRs* (%).

Gene Name	Identity (%)				
	*LstiPR1*	*LstiPR2*	*LstiPR3*	*LstiPR4*	*LstiPR5*
*LstiPR1*	100.00				
*LstiPR2*	60.80	100.00			
*LstiPR3*	33.75	33.90	100.00		
*LstiPR4*	33.20	30.52	35.96	100.00	
*LstiPR5*	30.01	29.91	37.10	53.87	100.00

## Data Availability

All the data and resources generated for this study are included in the article.

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
