# Peer review of "Functional Characterization of Pheromone Receptors in the Beet Webworm, Loxostege sticticalis (Lepidoptera: Pyralidae)"

_insects, 2023, doi:10.3390/insects14070584_

Round 1

Reviewer 1 Report (Previous Reviewer 1)

The manuscript is substantially improved after revisions, however several concerns must be addressed before this manuscript shall be deemed acceptable for publication. 

Critically, there were several examples identified where the incorrect reference was indicated. For example: 

Line 79. Ref 31. It is mentioned in the preceding statement about expressed PRs, but reference 31 is a study about general ORs in Drosophila. 

Line 324. Ref. 53. It is mentioned in the preceding statement about Lepidoptera, however reference 53 is a study about aphids. 

Line 327. Ref. 19. The preceding statement is about moths, however, reference 19 is a study about a mosquito. 

Line 349. Ref. 26. The preceding statement mentions Bombyx mori, however, reference 26 concerns Heliothis virescens. 

Line 359. Refs 52 and 53 are about different species than the species mentioned in the text, Proeulia auraria and Spodoptera litura. 

It is of critical importance that these errors and all others regarding usage of incorrect references are corrected. 

Furthermore, one previously suggested revision was not made, that found on lines 348-349 in the present manuscript regarding the statement about the study in Bombyx mori activating the behavior of L. sticticalis. 

Aside from that, several minor concerns exist that should be addressed. 

Abstract: 

Lines 32-33. "Five candidate PRs were deorphanized." 

This is incorrect usage of the word deorphanized. Saying deorphanized means you identified key ligand(s) for the receptor. In reality, in this study, only one candidate PR was deorphanized and found to be a functional PR. It would be better to say that the Five candidate PRs were functionally assayed. 

Introduction. 

Line 60. "are common in some moths". 

It should be the case that this is true in most moths, not just some moths, especially all the moth species for which the males respond to female-produced pheromones. 

Materials and Methods. 

Line 111. "gender".

It is incorrect to refer to gender for most insects. Gender is a social construct. It is better to say "sex" here in this case instead of gender. 

Line 125. "MAFFT". 

For this approach, please indicate which parameters were used for the alignment, similar to how the parameters were mentioned after this for the phylogenetic tree construction. 

Results and Figures. 

Figure 3. Previously it was requested to add an arrowhead to point to the response in Panel C. However this was not done. It seems that arrows were added to Panel A instead. Please consider to add an arrow to point to the response in Panel C. Furthermore, in Panel A, it is not clear what the traces are showing in the bottom of the panel. Please add a description of what is being shown here in the appropriate section of the figure legend. 

Line 321 and 322. L. sticticalis is not written in italics. Please ensure the species name is written in italics here and throughout the rest of the manuscript. 

Line 340. "Therefore, we comprehensively characterized the function of five candidate LstiPRs"

It is incorrect to say that you comprehensively characterized the function of 5 candidate PRs when you found only one ligand for one PR. It would be better to say that you assayed the response of 5 candidate PRs to known pheromone compounds. 

Poor quality of language is found throughout the manuscript, with numerous small but highly visible errors present, especially in newly added text. This manuscript must be revised for language before re-submission. 

Reviewer 2 Report (Previous Reviewer 3)

The logic flow of the manuscript has been improved in the new version and the majority of my comments have been taken into account. Nevertheless the manuscript needs important language corrections (see below).

The manuscript needs careful language revision. There are many spelling and grammar mistakes and species names need to be spelled in italics throughout.

Reviewer 3 Report (New Reviewer)

In this manuscript, Zhang et.al., focused on the peripheral mechanism by which sex pheromone is encoded by pheromone receptors (PR). Overall, the experiments were carried out well and their conclusion goes well with the results obtained. I have a few minor comments which will surely strengthen the quality of the paper. I request authors to address these concerns before I make my recommendation for publication.

1. I am not convinced on how the ‘B’ neuron has been shown in Figure 3. Given the noise in the single unit recordings, it could be masked in the basal noise as the spike amplitude is not as prominent as the ‘A’ neuron spike. Also, I request the authors show the frequency of both ‘A’ and ‘B’ neurons in the spontaneous firing. If possible, authors can show PSTH before and during the odor stimulation.

2. In Figure 3, authors showed the E11-14:OAc response using single unit recordings. The details were unclear to me. Please mention whether the spike frequency was calculated throughout the length of the stimulus. A PSTH will show what the response looks like. It would be nice to show the kinetics of the response.

3. The EAG response from the antennae has shown that there is a response for 14:OH (Figure 2). However, no response has been observed with that ligand when single unit recording was carried out. EAG is a change in field potential. Authors should discuss this discrepancy between EAG response and single unit recording in the discussion.

This manuscript is a resubmission of an earlier submission. The following is a list of the peer review reports and author responses from that submission.

Round 1

Reviewer 1 Report

The authors report on functional characterization of previously identified candidate pheromone receptors (PRs) of the beet webworm, Loxostege sticticalis. Antennal response to pheromone compounds is first confirmed by EAG then in individual olfactory neurons by SSR. Then five PR ORFs are cloned and co-expressed in standardly used Xenopus oocyte system, followed by functional assay with the pheromone compound. One PR is confirmed to respond to the major pheromone component while the remaining four did not respond to any of the compounds.

The manuscript is well written both in terms of language and in structure. Experiments are logically conceived and carried out. Minor revisions are suggested to improve the overall quality of the manuscript in terms of content and clarify.

Comments are provided by section:

Simple Summary/Abstract

Line 23-24. “The other four LstiPRs did not respond to to any sex pheromone compounds.” Here, and elsewhere, prior to functional assay, these PRs must be referred to as “candidate PRs” e.g. “the other four candidate LstiPRs.” Presumably, they are designated as PRs based upon phylogeny, but as they authors acknowledge at least one candidate PR does not respond to pheromones (CpomOR3), and there are likely to be others. Only after a candidate PR is confirmed to respond to pheromone can it be appropriately designated as a PR.

Line 33-35. “EAG assays showed that the major sex pheromone component elicited a strong antennae response in male moths, an that it was specifically recognized by the LstiPR2.” It is unclear here how the EAG assay would have shown that the pheromone was recognized by PR2. That knowledge did not come from the EAG but rather the Xenopus assay. So this section should be revised.

Introduction.

Pag3 2, Last Paragraph. This paragraph describes the various pheromone compounds of the beet webworm. It would be good for the reader if relevant text were included providing clarification on whether these pheromones are Type I, Type II or other. See Zhang and Lofstedt 2015, Frontiers in Ecology and Evolution

Line 98. Change “perception” to “detection”

Line 98-99. “Although the genes involved in pheromone perception have been identified, the function of LstiPRs remains unknonwn [31].” Reminder to refer to them here as candidate PRs, but also providing some more details supporting this statement, as these genes are central to this study. A couple of statements about what was found in [31], for example, was it a transcriptomic and genomic study, how many candidate PRs were identified, etc.

Materials and Methods.

Section 2.7. Xenopus Oocyte System.

Line 172-173. “A mixture of LstiPRs and LstiOrco cRNA (1:1) was injected…” This is unclear. As written it sounds like could have been the case that cRNA for all of candidate PRs was mixed together with Orco, when it is obvious that cRNAs for each candidate PR would have been individually mixed with cRNA of Orco. A simple revision is required to clarify this.

Line 180. “upon stimulation with the different pheromones.” Some additional text is required here indicating what exactly was done here, rather how it was done.

Results.

Line 219-220. “encoding polypeptides of 326,421,362,436, and 346 amino acids, respectively.” Some of these are atypically short for complete OR ORFs, which are typically in the range of 375-420. Especially the two that are 326 and 346. Is it confirmed that these are the actual full length ORFs for these genes? Since the authors have cloned out each of the ORFs for this report, the manuscript would benefit from inclusion of additional information on the clones that were used for Xenopus oocyte assays. At the very least, the cloned sequences should be included as supplemental info, and to provide greater confidence that the ORFs are complete, reference to the relevant genomic/transcriptomic accessions from [31] would be useful, and also providing info on predicted secondary structure in terms of membrane topography and predicted transmembrane regions.

Discussion.

Line 323-324. “the LstiPRs were narrowly tuned to the sex pheromone chemical in the Xenopus oocyte system.” It is inaccurate to say the LstPRs were narrowly tuned since only one of them responded to anything, and in this case, only pheromone compounds were assayed. Please revise this statement.

Line 327-329. “These results were inconsistent with the study done in Bombyx mori which found only one pheromone component activated the behavior of L. sticticalis.” This does not make sense. How does a study in B. mori show that one component activates the behavior of L. sticticalis?

Figures/Tables

Figure 4. In Figure 4c, it is not clear where the neuronal response is. It would be helpful to highlight the response, for example, with an arrowhead pointing to it.

Author Response

Point 1: Simple Summary/Abstract

Line 23-24. “The other four LstiPRs did not respond to to any sex pheromone compounds.” Here, and elsewhere, prior to functional assay, these PRs must be referred to as “candidate PRs” e.g. “the other four candidate LstiPRs.” Presumably, they are designated as PRs based upon phylogeny, but as they authors acknowledge at least one candidate PR does not respond to pheromones (CpomOR3), and there are likely to be others. Only after a candidate PR is confirmed to respond to pheromone can it be appropriately designated as a PR. 

Response 1: Thanks for your advise and you are very rigorously. We have revised the description before the PR identified the function in the manuscript.

Point 2: Line 33-35. “EAG assays showed that the major sex pheromone component elicited a strong antennae response in male moths, an that it was specifically recognized by the LstiPR2.” It is unclear here how the EAG assay would have shown that the pheromone was recognized by PR2. That knowledge did not come from the EAG but rather the Xenopus assay. So this section should be revised.

Response 2: Sorry for this mistake description. We have revised it in the revised manuscript.

Point 3: Introduction.

Pag3 2, Last Paragraph. This paragraph describes the various pheromone compounds of the beet webworm. It would be good for the reader if relevant text were included providing clarification on whether these pheromones are Type I, Type II or other. See Zhang and Lofstedt 2015, Frontiers in Ecology and Evolution

Response 3: Thanks for your advise and you are very professional. Indeed, the type of pheromone information is important, we have added the type information in revised manuscript. However, the reference “Zhang and Lofstedt 2015, Frontiers in Ecology and Evolution” was not index by the key word, we use another reference to support it. The reference is “Ando, T.; Inomata, S.I.; Yamamoto, M,. Lepidopteran sex pheromones. Top. Curr. Chem. 2004, 239, 51-96”.

Point 4: Change “perception” to “detection”

Response 4: Thank for your reminder. We have revised it in the manuscript.

Point 5:Line 19. Line 98-99. “Although the genes involved in pheromone perception have been identified, the function of LstiPRs remains unknonwn [31].” Reminder to refer to them here as candidate PRs, but also providing some more details supporting this statement, as these genes are central to this study. A couple of statements about what was found in [31], for example, was it a transcriptomic and genomic study, how many candidate PRs were identified, etc.

Response 5: Thanks for your reminder. We have revised the all text description about PRs in revised manuscript, according to your advise.

Point 6: Materials and Methods.

Section 2.7. Xenopus Oocyte System.

Response 6: Thanks your reminder. However, according to other articles, the initials of  Xenopus Oocyte System do not necessary capital letters.

Point 7: Line 172-173. “A mixture of LstiPRs and LstiOrco cRNA (1:1) was injected…” This is unclear. As written it sounds like could have been the case that cRNA for all of candidate PRs was mixed together with Orco, when it is obvious that cRNAs for each candidate PR would have been individually mixed with cRNA of Orco. A simple revision is required to clarify this.

Response 7: Thanks for your advise. We have revised it in the revised manuscript that the description is more clear.

Point 9: Line 180. “upon stimulation with the different pheromones.” Some additional text is required here indicating what exactly was done here, rather how it was done.

Response 8: Sorry for this unclear description, we have add more details description in the revised manuscript.

Point 9: Results.

Line 219-220. “encoding polypeptides of 326,421,362,436, and 346 amino acids, respectively.” Some of these are atypically short for complete OR ORFs, which are typically in the range of 375-420. Especially the two that are 326 and 346. Is it confirmed that these are the actual full length ORFs for these genes? Since the authors have cloned out each of the ORFs for this report, the manuscript would benefit from inclusion of additional information on the clones that were used for Xenopus oocyte assays. At the very least, the cloned sequences should be included as supplemental info, and to provide greater confidence that the ORFs are complete, reference to the relevant genomic/transcriptomic accessions from [31] would be useful, and also providing info on predicted secondary structure in terms of membrane topography and predicted transmembrane regions.

Response 9: Thanks for your reminder. Usually, the length of OR range 375-420, however, some genes are short and the transmembrane regions were not predicted precisely by present tools. In this studies, we use the technique rapid-amplification of cDNA ends (RACE) to cloned the candidate PRs, thus, in the absence of genomic data, we can think it is the candidate PRs full-length, presently. Indeed, it is possible that 326 and 346 are not full-length, because we cannot detect the function of this two candidate. We do not exclude this possibility, but merely speculate. Therefore,  we will sequencing the whole genome of L. sticticalis to verity it in the future.

Point 10: Discussion.

Line 323-324. “the LstiPRs were narrowly tuned to the sex pheromone chemical in the Xenopus oocyte system.” It is inaccurate to say the LstPRs were narrowly tuned since only one of them responded to anything, and in this case, only pheromone compounds were assayed. Please revise this statement.

Response 10: Thanks for your reminder. We have revised it in manuscript.

Point 11: Line 327-329. “These results were inconsistent with the study done in Bombyx mori which found only one pheromone component activated the behavior of L. sticticalis.” This does not make sense. How does a study in B. mori show that one component activates the behavior of L. sticticalis?

Response 11: Sorry for this mistake about my careless. We have revised description in the manuscript.

Point 12: Figures/Tables

Figure 4. In Figure 4c, it is not clear where the neuronal response is. It would be helpful to highlight the response, for example, with an arrowhead pointing to it.

Response 12: Thanks for your advise. We have revised it in manuscript.

Reviewer 2 Report

To assess the importance of this work, the authors need to present the methods and sequence of the study more fully and reasonably.

Please specify the timing of the experiments. Were all experiments carried out using one laboratory generation or were the insects reproduced in the laboratory for several rounds? Indicate the number of insects used in each experiment, as well as the number of repetitions of each experiment.

There are no references to electroantennography responses technique (2.6). If this technique was developed by the authors of this manuscript, it should be justified. Is it acceptable to offer several (six) pheromone components to one male at short time intervals (30 s)? How can you explain the weak reaction of the male to the pheromone compound that have been identified in L. sticticalis, except as a violation of the purity of the experiment?

Information should be added to Table 2 on who and when identified each pheromone compound.

Line 19. “insect pest that is found worldwide” – This is an exaggeration.

Line 53. [6], ,

Line 92. Add more recent references. The reference [29] does not contain information on the distribution of the beet webworm.

Line 107. It is necessary to clarify the time of collection and the stage (cocoons, adults or larvae (instar)) of the collected insects, as well as the number of insects.

Lines 288-294. This paragraph should be moved to the introduction.

Lines 327-331. The meaning of the sentence is not clear.

Line 336. L. sticsticalis – typo.

Author Response

Point 1: To assess the importance of this work, the authors need to present the methods and sequence of the study more fully and reasonably. 

Response 1: Thanks for your suggestion. The methods were given more details in the revised manuscript, and the sequence of the pheromone receptor were given in the supplementary material.

Point 2: Please specify the timing of the experiments. Were all experiments carried out using one laboratory generation or were the insects reproduced in the laboratory for several rounds? Indicate the number of insects used in each experiment, as well as the number of repetitions of each experiment.

Response 2: The insect in this study was used one laboratory generation. In detail, we collected the larvae from filed and rared in the laboratory, than the larvae pupated and feathered. The antennae of three-old-day male adults were used in this article. And the timing of the experiments was added details in revised manuscript.

Point 3: There are no references to electroantennography responses technique (2.6). If this technique was developed by the authors of this manuscript, it should be justified. Is it acceptable to offer several (six) pheromone components to one male at short time intervals (30 s)? How can you explain the weak reaction of the male to the pheromone compound that have been identified in L. sticticalis, except as a violation of the purity of the experiment?

Response 3: Sorry for my mistake about the method describe. The EAG assay was according the article “Electrophysiological Measurements from a Moth Olfactory System”, which was published by Journal of Visualized Experiments. The interval time is a writing mistake, and it is should 3 min. This writing mistake have revised in manuscript.

Point 4: Information should be added to Table 2 on who and when identified each pheromone compound.

Response 4: Thank for your reminder. The information of pheromone compounds in L. sticticalis were identified by Prof. Ai-ping Liu and the reference article have added footnote below the Table 2 in the revised manuscript.

Point 5:Line 19. “insect pest that is found worldwide” – This is an exaggeration.

Response 5: Thanks for your reminder for this exaggeration describe. We have revised the describe in revised manuscript-” insect pest that is found in Russia, Canada, Britain, and China”.

Point 6: Line 53. [6], Line 92. Add more recent references. The reference [29] does not contain information on the distribution of the beet webworm.

Response 6: Sorry for my mistake. We have added the more recent references in the revised manuscript.

Point 7:Line 107. It is necessary to clarify the time of collection and the stage (cocoons, adults or larvae (instar)) of the collected insects, as well as the number of insects.

Response 7: Thanks for your reminder. The insect of this article were collected the stage range two to five instar that more than 1000 individuals, and the collection time of insect in June, 2021. More detail describe was added in the revised manuscript.

Point 9: Lines 288-294. This paragraph should be moved to the introduction.

Response 8: Thanks for your suggestion, we have moved this paragraph to the part of introduction.

Point 9: Lines 327-331. The meaning of the sentence is not clear.

Response 9: Sorry for this not clear describe. We have revised the describe in the manuscript.

Point 10: Line 336. L. sticsticalis– typo.

Response 10: Sorry for this mistake. The correct word spelling was revised in manuscript.

Reviewer 3 Report

The manuscript presents a large amount of data with classical approaches and most parts seem to be well performed. The manuscript needs, however, thorough revision before it can be considered for publication.

Especially stream-lining of introduction and discussion would be important, the material and methods section lacks some important information, and some clarifications in the result part would be helpful for the reader.

A first fundamental question is whether there is any way to compare stimulus doses used in electrophysiological recordings with doses used in oocyte patch clamp recordings? It is maybe not so surprising that you didn’t get any response when expressing other PRs than LstiPR2 in oocytes, if you need potentially very high doses to elicit an antennal response. The same is true for the absence of SSR responses to the other pheromone compounds. If you don’t get a response in EAGs, why would you expect a response to the same dose in single sensilla? And of course the different PRs might be expressed in different sensillum types…

Please find detailed comments below.

Simple Summary:

I am not sure that this corresponds well to what is asked for in the simple summary. I would try to formulate the obtained results in a more general way. And maybe be a bit more cautious with announcing that your study will contribute to improve environment-friendly pest control.

Abstract:

Last sentence of the abstract is unclear if not misleading: you did not work on detection of interspecific signals, the only discussion is about other species using the same compound as the main compound in your species.

Introduction:

Second paragraph: You start with a general description of Insect antennae. If you introduce the different parts of the antenna, you have to mention that you talk afterwards only about the flagellum (and in some insects even the gross structure of the antennae can be different, e.g. in flies). Also there are many more sensilla types in insects than those you list. Either be more specific and talk about moths, or don’t list all sensillum types, but only introduce the potentially olfactory types?

Lines 74-75: weird formulation: the heteromeric complexes allow activation of the ion channels, whereas regulation of mating behavior is occurring through central processing of pheromone information.

Line 78: mention that this work has been done on PRs + a co-receptor?

Lines 85-88: the two sentences are somewhat redundant

Line 97: An information of crucial importance is missing here: what are behavioral responses of males to individual compounds or mixtures? Apparently wind tunnel experiments have been performed, but the Chinese journal in which this is published is not easily accessible internationally, so some more details should be given. Also, if a mixture of 3 compounds at the indicated ratio is effective in the field, probably larger amounts of E11-14:OH are necessary and should potentially have been used in electrophysiological experiments?

Last paragraph of the introduction, introducing the aims of the study:

What is the added value of EAGs? Mention the EAG experiments in this part of your introduction and why they were performed

Last sentence of introduction: You say that you confirm oocyte results with SSR recordings. But how can you correlate SSR responses with PR receptor expression without in situ hybridization of the receptor and identifying the corresponding sensilla? In my eyes, the only thing you can say from your SSR experiments is that most likely LstiPR2 is expressed in b neurons of the type of trichoid sensilla you recorded from.

Materials and Methods:

Part 2.1: were pupae sexed? If not, how did you ensure that males had not mated before?

The last sentence should probably transfered to section 2.2 (or otherwise other information on insects used for electrophysiology experiments should be provided)

Part 2.5: mention for what these dilutions were used (oocyte recordings?)

Line 165: please clarify: stimulated with each pheromone compound before and after what? Did you alternate between paraffin oil and pheromone compounds?

Part 2.7: please specify the pheromone dilutions used for stimulation (is this what you describe in 2.1?) and also mention how you applied the stimulation solutions?

Part 2.8: please specify what type of electrodes you used

Lines 203-204: please be more precise in the description of your data analysis: did you subtract spontaneous activity from the response after stimulation? In this case during which period of time?

Results:

Figure 2: in a) can you add paraffin oil responses (unless you subtracted them from your pheromone responses)? In b I would call this rather EAG traces. Please also mention in the legend the dose used for stimulation (even if it is mentioned already in the text): this is really important, because you use a pretty high dose of pheromone, but apparently you would need even higher doses to obtain responses to the other compounds (which must nevertheless be detected, because they are behaviorally active as far as I understand).

Figure 3c: The lack of response for other PR/Orco combinations can either tell you that the expression was not functional or that the receptors did not respond to the stimuli at the doses you tested. I suggest to discuss this somewhere.

Part 3.4.: What is the percentage of trichoid sensilla in which you found a b neuron responding to E11-14:OAc? In all? Or just a certain proportion regardless of the position of the sensillum? Or in trichoid sensilla at a specific part of the antenna?

Figure 4: your spike traces are not terribly convincing…. With a large noise level, I am not sure how you were able to count the numbers of small amplitude spikes. It would be helpful to show an example with high resolution which indicates the spikes you counted (e.g. with dots as in 4A?).

Discussion:

Lines 312-314: the references are not correct, unless you change the sentence to something like: …that recognize these five pheromone components, as shown in other Lepidopteran species.

Lines 322-324: Your sentence is not really correct. You only show that PR2 might be narrowly tuned, you don’t know about the others, as you did not get any response.

Lines 327-331: this sentence is incomprehensible and this part needs to be clarified: you compare your species with findings in other species. If you want to compare correctly you need behavioral responses of your species: do males respond to the major compound alone (like in Bombyx) or not? The comparison with Cydia is risky, because pear ester is a plant-emitted volatile, which also serves as a pheromone.

Lines 344-346: This is important: 1. You might not have found the sensilla trichoidea detecting the other compounds, especially if there are only few, 2. The doses you used for stimulation might very likely have been too low, 3. and indeed other types of sensilla might be involved in the detection (but these also contribute to EAG responses, so again the dose-problem is the same).

Conclusions:

Indeed other species are using the same pheromone compound, but what is important for species recognition is generally the blend of compounds in the correct ratio: for this peripheral detection of all compounds involved is necessary, but the recognition of the correct blend is happening in the brain. So, as mentioned before it is crucial to know what males of your species respond to behaviorally: only the major compound, or specific blends? The field data you mention seem to indicate that a 3-component blend seems to be more efficient than the major compound?

Language/grammar corrections

Line 27 : …communication….that are expressed in…

Lines 28-29: …in this sexual communication.

Lines 30-31: …about sexual communication in…

Line 34: … a strong antennal response…

Line 41: …interspecific communication…

Line 52: …in Bombyx mori (6), …

Line 66: Peripheral detection of sex pheromones…

Line 68: …degrading sex pheromones.

Lines 71-72: …the main ligand-gated ion channels

Line 88: ..on its dendritic membrane…

Line 164: with an inter-stimulus interval of 30 s.

Line 186: Single sensillum recordings…

Line 240: … elicited by the pheromone compound…

Line 241: …higher than the response to the other pheromone compounds at the same dose…

Line 244: …weak EAG responses similarly to paraffin oil?

Lines 253-254: …which is the main pheromone compound of…

Line 259: …at the highest tested dose?

Line 274: …distinguished through the spike amplitudes….

Line 310: .. an induction effect on male behavior in the wind tunnel….

Line 324: …was specifically responding to…

Line 332: …are expressed on the dendrites of ORNs within antennal sensilla,…

Line 338: …that trichoid sensilla (or: sensilla trichoidea)…

Author Response

Point 1: The manuscript presents a large amount of data with classical approaches and most parts seem to be well performed. The manuscript needs, however, thorough revision before it can be considered for publication. Especially stream-lining of introduction and discussion would be important, the material and methods section lacks some important information, and some clarifications in the result part would be helpful for the reader. A first fundamental question is whether there is any way to compare stimulus doses used in electrophysiological recordings with doses used in oocyte patch clamp recordings? It is maybe not so surprising that you didn’t get any response when expressing other PRs than LstiPR2 in oocytes, if you need potentially very high doses to elicit an antennal response. The same is true for the absence of SSR responses to the other pheromone compounds. If you don’t get a response in EAGs, why would you expect a response to the same dose in single sensilla? And of course the different PRs might be expressed in different sensillum types…

Response 1: Thanks for your approve of this article. The question that whether there is any way to compare stimulus doses used in electrophysiological recordings with doses used in oocyte patch clamp recordings. The answer is NO. Because of three points: 1. concentration units are different, 2. the assays conditions absolutely different, thus, there is no comparability, 3. the EAG is test the whole antennae response of odors, however, the Xenopus oocyte system only test response that one odor to one receptor. In this study, we just initial research the molecular mechanism in the laboratory condition. Indeed, it is not so surprising that the other PRs do not elicit by pheromones, however, the EAG low or not response not represent the SSR can’t test the response, because of the individual sensilla has more sensitive. In the SSR assays, we want to identified the pheromone response ORNs, so we test the all pheromone again. Nevertheless, the more higher concentration of the pheromones were not necessary in this study. The PR might expressed in different sensillum types, it is possible, we have discussion it in the manuscript, this part we will deeper analysis in the future.

Point 2: 

Simple Summary:

I am not sure that this corresponds well to what is asked for in the simple summary. I would try to formulate the obtained results in a more general way. And maybe be a bit more cautious with announcing that your study will contribute to improve environment-friendly pest control.

Response 2: Thanks for your advise. We have revised our results description, cautiously.

Point 3:

Abstract:

Last sentence of the abstract is unclear if not misleading: you did not work on detection of interspecific signals, the only discussion is about other species using the same compound as the main compound in your species.

Response 3: Sorry due to our unclear describe misleading for reader. The last sentence in the part of abstract may describe not appropriate. We have revised it in the manuscript.

Point 4: .

Introduction:

Second paragraph: You start with a general description of Insect antennae. If you introduce the different parts of the antenna, you have to mention that you talk afterwards only about the flagellum (and in some insects even the gross structure of the antennae can be different, e.g. in flies). Also there are many more sensilla types in insects than those you list. Either be more specific and talk about moths, or don’t list all sensillum types, but only introduce the potentially olfactory types?

Response 4: Thanks for your advise. Indeed, the insect sensilla type are more than we list that is our carelessness. Thus, we narrowed the insect range to Lepidoptera. The large numbers research clarify that the sensilla trichodea are mainly recognize the pheromone compounds from females moth in Lepidoptera. Consequently, we mainly aiming to analysis the molecular mechanism that how the male recognize the pheromone compounds.

Point 5: Lines 74-75: weird formulation: the heteromeric complexes allow activation of the ion channels, whereas regulation of mating behavior is occurring through central processing of pheromone information.

Response 5: Sorry for this weird formulation due to my poor English. We have revised describe about the odor molecular mechanism by odor recognize in manuscript.

Point 6: Line 78: mention that this work has been done on PRs + a co-receptor?

Response 6: This mention just introduce the background and molecular mechanism about pheromone receptor (PRs) for this article, which can provide the research approach of PRs in L. sticticalis. Also, maybe our describe unclear lead the misunderstanding, thus we have revised describe in manuscript.

Point 7: 85-88: the two sentences are somewhat redundant

Response 7: Thanks for your suggestion. We have delete these two sentences in revised manuscript.

Point 8: Line 97: An information of crucial importance is missing here: what are behavioral responses of males to individual compounds or mixtures? Apparently wind tunnel experiments have been performed, but the Chinese journal in which this is published is not easily accessible internationally, so some more details should be given. Also, if a mixture of 3 compounds at the indicated ratio is effective in the field, probably larger amounts of E11-14:OH are necessary and should potentially have been used in electrophysiological experiments?

Response 8: The behavioral responses of males to individual compounds or mixtures have analysis in the previous studies, and the reference article is in the list of manuscript reference part Liu et al., 2011. This research just research the molecular mechanism of PRs recognize the pheromone compounds in L. sticticalis. The results of wind tunnel experiments published in Chinese journal, we have given more details in revised manuscript. The concentrate of E11-14:OH in electrophysiological experiments is too high, if we use a more higher concentrate, it is nonconformity natural concentrate. In this research, the pheromone compound of E11-14:OH did not elicit the response of LstiPRs, it may be due to the separate organ. Furthermore, the EAG test low response or not, it is not represent this compounds not regulate the behaviors in insect. However, in the laboratory test, this concentrate is enough.

Point 9: Last paragraph of the introduction, introducing the aims of the study:

What is the added value of EAGs? Mention the EAG experiments in this part of your introduction and why they were performed

Last sentence of introduction: You say that you confirm oocyte results with SSR recordings. But how can you correlate SSR responses with PR receptor expression without in situ hybridization of the receptor and identifying the corresponding sensilla? In my eyes, the only thing you can say from your SSR experiments is that most likely LstiPR2 is expressed in b neurons of the type of trichoid sensilla you recorded from.

Response 9: You are very professional. The EAG experiments in this part, the reason is this assays can certify that the antennae are the mainly organ of pheromone compounds recognize in L. stiticalis. Indeed, the experiment of in situ hybridization are not improved, our describe is more absolutely, we have revised it in manuscript.

Point 10: Materials and Methods:

Part 2.1: were pupae sexed? If not, how did you ensure that males had not mated before?

Response 10: Your question is very interest. We can through the end of pupae to distinguish the sex, easily.

Point 11: The last sentence should probably transfered to section 2.2 (or otherwise other information on insects used for electrophysiology experiments should be provided)

Response 11: Thanks for your advise, and we have transferred it in the revised manuscript.

Point 12: Part 2.5: mention for what these dilutions were used (oocyte recordings?)

Response 12: Yes. This dilutions were used in the Xenopus oocyte system, and we have added it in the revised manuscript.

Point 13: Line 165: please clarify: stimulated with each pheromone compound before and after what? Did you alternate between paraffin oil and pheromone compounds?

Response 13: The eclectic signals values variation difference that before and after each pheromone compound stimuli were recorded. The average values of paraffin oil that stimuli before and after each pheromone compound as negative control.

Point 14: Part 2.7: please specify the pheromone dilutions used for stimulation (is this what you describe in 2.1?) and also mention how you applied the stimulation solutions?

Response 14: Sorry, it is my carelessness. The details concentration of pheromone compounds were added in the revised manuscript.

Point 15: Part 2.8: please specify what type of electrodes you used. Lines 203-204: please be more precise in the description of your data analysis: did you subtract spontaneous activity from the response after stimulation? In this case during which period of time?

Response 15: The tungsten microelec trodes were used in this part, and have added in the revised manuscript. And the response formulate were base on the numbers of spikes of 1s before and after that stimulate, and more description were added in the revised manuscript.

Point 16: Results:

Figure 2: in a) can you add paraffin oil responses (unless you subtracted them from your pheromone responses)? In b I would call this rather EAG traces. Please also mention in the legend the dose used for stimulation (even if it is mentioned already in the text): this is really important, because you use a pretty high dose of pheromone, but apparently you would need even higher doses to obtain responses to the other compounds (which must nevertheless be detected, because they are behaviorally active as far as I understand).

Response 16: Thanks for your suggestion. We have added the details about concentration of stimulate pheromone compounds and the pheromone response was subtracted the paraffin oil response.  

Point 17: Figure 3c: The lack of response for other PR/Orco combinations can either tell you that the expression was not functional or that the receptors did not respond to the stimuli at the doses you tested. I suggest to discuss this somewhere.

Response 17: Thanks for your advise, in fact, we have discuss this in the paragraph 5 last sentence of discussion that is description unclear. We have revised it in the manuscript.

Point 18: Part 3.4.: What is the percentage of trichoid sensilla in which you found a b neuron responding to E11-14:OAc? In all? Or just a certain proportion regardless of the position of the sensillum? Or in trichoid sensilla at a specific part of the antenna?

Response 18: We choice the sensilla trichoid in antennae’s different sites are randomly. All the results showed that the b neuron was elicit by E11-14:OAc in this research. Maybe when we test the larger numbers of sensilla trichoid, it is possible that we can identify the other sub-type. However, the percentage may be very lower.

Point 19: Figure 4: your spike traces are not terribly convincing…. With a large noise level, I am not sure how you were able to count the numbers of small amplitude spikes. It would be helpful to show an example with high resolution which indicates the spikes you counted (e.g. with dots as in 4A?).

Response 19: Thanks for your advise. Although the assays with larger noise level, the response of pheromone are very obliviously. And the numbers of spikes was count by profession analysis software, not by artificial. Therefore, It is not necessary to improved with high resolution.

Point 20: Discussion:

Lines 312-314: the references are not correct, unless you change the sentence to something like: …that recognize these five pheromone components, as shown in other Lepidopteran species.

Response 20: Thanks for your suggestion, and we have revised it in manuscript.

Point 21: Lines 322-324: Your sentence is not really correct. You only show that PR2 might be narrowly tuned, you don’t know about the others, as you did not get any response.

Response 21: Sorry, it is our un-rigorously. We have qualify the conditions in the revised manuscript, aiming the description more cautionary.

Point 22: Lines 327-331: this sentence is incomprehensible and this part needs to be clarified: you compare your species with findings in other species. If you want to compare correctly you need behavioral responses of your species: do males respond to the major compound alone (like in Bombyx) or not? The comparison with Cydia is risky, because pear ester is a plant-emitted volatile, which also serves as a pheromone.

Response 22: In this part, we want to expression two aspects. One aspect is LstiPR2 can elicit by pheromone E11-14:OAc, that the recognize molecular mechanism similar to the Bombyx mori, which is elicit by one pheromone. Another aspect is the non-functional LstiPRs (LstiPR1, LstiPR3, LstiPR4, and LstiPR5) may not take part in the pheromone recognize, it may be elicit by other odors, such as host volatile. Maybe our description unclear, we have revised it in the manuscript.

Point 23: Lines 344-346: This is important: 1. You might not have found the sensilla trichoidea detecting the other compounds, especially if there are only few, 2. The doses you used for stimulation might very likely have been too low, 3. and indeed other types of sensilla might be involved in the detection (but these also contribute to EAG responses, so again the dose-problem is the same).

Response 23: We are very agree with your statement. Maybe when the larger number of sensilla trichoidea were test, the other compounds can be test the sub-type, however, the percentage may very low. Also, it is necessary that deeper analysis this in the future. The dose we used compared with the natural is too higher, thus the more higher dose is not necessary, because the olfactory is multiple, may be this behaviour regulate by other feeling system, such as visual or touch system. In addition, the low responses were not represent the low regulate behavior. We just control unique variables to test it that is not reaction the true fact.

Point 24: Conclusions:

Indeed other species are using the same pheromone compound, but what is important for species recognition is generally the blend of compounds in the correct ratio: for this peripheral detection of all compounds involved is necessary, but the recognition of the correct blend is happening in the brain. So, as mentioned before it is crucial to know what males of your species respond to behaviorally: only the major compound, or specific blends? The field data you mention seem to indicate that a 3-component blend seems to be more efficient than the major compound?

Response 24: Your suggestion is certainly significant for us, and we will deeper research it in the future. In the previously study, the research provide that 3-component blend was the optimal ratio, however, this ratio was the artificial choice it, randomly. What the pheromone ration release of female L. sticticalis in natural and how to distinguish the inter-/intra-specific pheromone components with other species are still unclear. In this study, we only initial explore the molecular recognition mechanism of individual sex pheromones of single PR pairs in L. sticticalis. Next, will use GC-MS to identified the natural ratio of female L. sticticalis, and performed the SSR, and wind tunnel assays to analysis the pheromones mixtures and analogs periphery peripheral recognition mechanism. The aim is to screen a optimal pheromones mixture ratio or synergistic, which increases the trapping of pheromone trapping products.

Point 25: Language/grammar corrections

Line 27: …communication….that are expressed in…

Lines 28-29: …in this sexual communication.

Lines 30-31: …about sexual communication in…

Line 34: … a strong antennal response…

Line 41: …interspecific communication…

Line 52: …in Bombyx mori (6), …

Line 66: Peripheral detection of sex pheromones…

Line 68: …degrading sex pheromones.

Lines 71-72: …the main ligand-gated ion channels

Line 88: ..on its dendritic membrane…

Line 164: with an inter-stimulus interval of 30 s.

Line 186: Single sensillum recordings…

Line 240: … elicited by the pheromone compound…

Line 241: …higher than the response to the other pheromone compounds at the same dose…

Line 244: …weak EAG responses similarly to paraffin oil?

Lines 253-254: …which is the main pheromone compound of…

Line 259: …at the highest tested dose?

Line 274: …distinguished through the spike amplitudes….

Line 310: .. an induction effect on male behavior in the wind tunnel….

Line 324: …was specifically responding to…

Line 332: …are expressed on the dendrites of ORNs within antennal sensilla,…

Line 338: …that trichoid sensilla (or: sensilla trichoidea)

Response 25: Thanks for your indicated. We have correct it all in the revised manuscript. And we find a professional company, which the mother language is English to revised language and grammar for full text.

Round 2

Reviewer 2 Report

I think that the authors failed to significantly improve the manuscript.

I asked “to present the sequence of the study more fully and reasonably”, but authors wrote about the sequence of the pheromone receptor.

There are no answers to the following questions: Is it acceptable to offer several (six) pheromone components to one male? How can you explain the weak reaction of the male to the pheromone compound that have been identified in L. sticticalis, except as a violation of the purity of the experiment?

Response 5. The new version of the phrase is just as unacceptable as the previous one.

Response 9. Lines 345-349. The new version of the sentence did not add clarity.

Reviewer 3 Report

Thank you for your efforts to take my comments into consideration. Unfortunately you dod not adequately address many of the comments and during re-writing you introduced new problems. I don't know if this is due to language problems. I have the impression that you misunderstood many of my suggestions and your explanations are unfortunately often very difficult to understand.